# Elucidating Scent and Color Variation in White and Pink-Flowered *Hydrangea arborescens* ‘Annabelle’ Through Multi-Omics Profiling

**DOI:** 10.3390/plants15010155

**Published:** 2026-01-04

**Authors:** Yanguo Ke, Dongdong Wang, Zhongjian Fang, Ying Zou, Zahoor Hussain, Shahid Iqbal, Yiwei Zhou, Farhat Abbas

**Affiliations:** 1Yunnan Urban Agricultural Engineering & Technological Research Center, College of Economics and Management, Kunming University, Kunming 650208, China; keyanguo@kmu.edu.cn (Y.K.);; 2Guangdong Provincial Key Laboratory of Ornamental Plant Germplasm Innovation and Utilization, Environmental Horticulture Research Institute, Guangdong Academy of Agricultural Sciences, Guangzhou 510640, China; 3Institute of Tropical Fruit Trees, Hainan Academy of Agricultural Sciences/Key Laboratory of Genetic Resources Evaluation and Utilization of Tropical Fruits and Vegetables (Co-Construction by Ministry and Province), Ministry of Agriculture and Rural Affairs/Key Laboratory of Tropical Fruit Tree Biology of Hainan Province, Haikou 571100, China; 4College of Horticulture, South China Agricultural University, Guangzhou 510642, China; 5Department of Horticulture, Faculty of Agriculture, Ghazi University, Dera Ghazi Khan 32200, Pakistan; 6Department of Horticulture, College of Agriculture, University of Sargodha, Sargodha 40100, Pakistan

**Keywords:** hydrangea, floral aroma, floral color, UPLC–MS/MS metabolomic, transcriptional regulation

## Abstract

The color and scent of flowers are vital ornamental attributes influenced by a complex interaction of metabolic and transcriptional mechanisms. Comparative analyses were performed to determine the molecular rationale for these features in *Hydrangea arborescens*, between the white-flowered variety ‘Annabelle’ (An) and its pink-flowered variant ‘Pink Annabelle’ (PA). Gas chromatography–mass spectrometry (GC–MS) detected 25 volatile organic compounds (VOCs) in ‘An’ and 21 in ‘PA’, with 18 chemicals common to both types. ‘An’ exhibited somewhat higher VOC diversity, whereas ‘PA’ emitted much bigger quantities of benzenoid and phenylpropanoid volatiles, including benzaldehyde, benzyl alcohol, and phenylethyl alcohol, resulting in a more pronounced floral scent. UPLC–MS/MS metabolomic analysis demonstrated obvious clustering of the two varieties and underscored the enrichment of phenylpropanoid biosynthesis pathways in ‘PA’. Transcriptomic analysis revealed 11,653 differentially expressed genes (DEGs), of which 7633 were elevated and linked to secondary metabolism. Key biosynthetic genes, including *PAL*, *4CL*, *CHS*, *DFR*, and *ANS*, alongside transcription factors such as *MYB*—specifically *TRINITY_DN5277_c0_g1*, which is downregulated in ‘PA’ (homologous to *AtMYB4*, a negative regulator of flavonoid biosynthesis)—and *TRINITY_DN23167_c0_g1*, which is significantly upregulated in ‘PA’ (homologous to *AtMYB90*, a positive regulator of anthocyanin synthesis), as well as *bHLH*, *ERF*, and *WRKY* (notably *TRINITY_DN25903_c0_g1*, highly upregulated in ‘PA’ and homologous to *AtWRKY75*, associated with jasmonate pathway), demonstrating a coordinated activation of color and fragrance pathways. The integration of metabolomic and transcriptome data indicates that the pink-flowered ‘PA’ variety attains its superior coloring and aroma via the synchronized transcriptional regulation of the phenylpropanoid and flavonoid pathways. These findings offer novel molecular insights into the genetic and metabolic interplay of floral characteristics in Hydrangea.

## 1. Introduction

Floral color and scent are essential features that arise from complex interactions between metabolic processes and gene expression mechanisms. These attributes facilitate plant-pollinator interactions, affect reproductive success, and augment esthetic value in horticulture [1,2,3]. The interaction between pigment biosynthesis and volatile compound production predominantly determines the phenotypic variation seen in angiosperms [4,5,6]. Consequently, comprehending the molecular regulation of these characteristics has emerged as a primary emphasis in plant molecular biology and breeding.

Flower coloration results from the synchronized manufacture, transport, and storage of pigments, primarily anthocyanins, flavonols, carotenoids, and chlorophyll derivatives within epidermal cells. Anthocyanins, derived from the flavonoid pathway, exhibit remarkable versatility, producing colors ranging from red to blue depending on pH, co-pigments, and the presence of metal ions [7,8,9]. The anthocyanin biosynthesis pathway is extensively conserved throughout plant species. Phenylalanine serves as the primary precursor for anthocyanin and flavonoid biosynthesis, undergoing catalysis by phenylalanine ammonia-lyase (PAL), cinnamate 4-hydroxylase (C4H), 4-coumarate CoA ligase (4CL), chalcone synthase (CHS) [10], and chalcone isomerase (CHI) to produce colorless naringenin [6,11,12]. Subsequently, colorless anthocyanins are generated from naringenin through the action of flavonoid 3-hydroxylase (F3H), flavonoid 3′-hydroxylase (F3′H), flavonoid 3′,5′-hydroxylase (F3′5′H), and dihydroflavonoid 4-reductase (DFR) [12,13,14]. Colorless anthocyanins may serve as intermediates in the formation of pigmented anthocyanins. Enzymatic processes, including those mediated by DFR and anthocyanin synthetase (ANS), transform these colorless forms into their pigmented equivalents. Colorless anthocyanins are generally present within a pH range of about 4 to 6, during which they lack the distinctive colors typical of anthocyanins [15]. Ultimately, the unstable anthocyanins are transformed into stable anthocyanin glycosides by ANS and UDP-glucose-flavonoid 3-*O*-glucosyltransferase (UFGT), exhibiting hues of red, pink, blue, and purple, among others, governed by the MYB–bHLH–WD40 transcriptional complex [16,17]. The genus *Hydrangea* (*Hydrangeaceae*) is renowned for its striking inflorescences, notable color polymorphism, and, in certain species, unique scent. These attributes render it an excellent material for studying the coordinated genetic and metabolic regulation of floral characteristics. Research on *H. macrophylla* indicates that aluminum ion complexation and soil pH markedly affect anthocyanin accumulation and sepal coloration [18,19]. Nonetheless, the precise transcriptional and metabolic pathways responsible for color production in *H. arborescens* are still insufficiently comprehended.

Floral aroma is triggered by the emission of volatile organic compounds (VOCs), which are mostly terpenoid, benzenoid/phenylpropanoid, and fatty acid derivatives [20,21]. Volatile terpenoids, primarily isoprene (C5), monoterpenes (C10), sesquiterpenes (C15), and diterpenes (C20), are the most prevalent group of plant volatile substances generated from the methylerythritol phosphate (MEP) and mevalonic acid pathways [22,23]. Phenylpropanoid volatiles are produced, encompassing phenylacetaldehyde, acetophenone (AP), 1-phenylethanol (1-PE), 2-phenylethanol (2-PE), and 2-phenylacetate. Additional benzenoid volatiles are produced from phenylalanine (Phe) through pathways other than cinnamate, including benzaldehyde (Bald), benzyl alcohol [24], benzyl acetate, and methyl salicylate (MeSA) [25,26,27]. These chemicals perform important ecological activities such as pollinator attraction, defense, and inter-plant communication. Floral volatile biosynthesis encompasses a number of enzymatic pathways and is heavily influenced by developmental, environmental, and genetic variables. In model plants such as *Petunia hybrida* and *Rosa rugosa*, important regulators such as ODORANT1 and EOBI have been identified as central to the generation of volatiles [28,29,30]. Despite these developments, little is understood about the genetic and biochemical regulation of VOC emissions in woody ornamentals such as *Hydrangea*.

Emerging evidence indicates that the pigment and volatile biosynthetic pathways share common precursors within the phenylpropanoid shikimate network, suggesting potential coordination between visual and olfactory traits [6,21]. Transcription factors that govern flavonoid biosynthesis may also affect VOC generation, forming integrated regulatory networks that dictate floral phenotype. Molecular interactions have been documented in several ornamental plants, such as *Petunia* [20,31], *Rosa* [32], and *Lilium* [33,34,35], but remain unexplored in *Hydrangea*. A combined transcriptome–metabolome strategy provides a thorough method to elucidate these connections by correlating gene expression with metabolite accumulation and pathway enrichment.

*H. arborescens*, also known as smooth hydrangea, is a deciduous shrub indigenous to eastern North America, extensively cultivated for its thick, dome-shaped flower clusters and tolerance to various environments [36,37]. The cultivar *H. arborescens* ‘Annabelle’ is notable for its unique white sepals at full bloom, which transition to green upon floral senescence. Previously, we performed an integrated metabolomic and transcriptome investigation of *H. arborescens* ‘Annabelle’ across floral developmental stages, discovering 33 volatile chemicals and more than 17,000 differentially expressed genes linked to floral smell production [18]. Moreover, comparative metabolomic analyses have indicated that flavonoid and phenylpropanoid derivatives are the principal factors influencing floral color intensity in *H. macrophylla* cultivars [19]. Notably, the pink-flowered ‘Pink Annabelle’ (PA) exhibits pink sepals at full bloom and possesses a more pronounced fragrance than ‘Annabelle’. This color variation is governed by complex metabolic networks involving anthocyanin biosynthesis and other pigment-related processes [38,39]. Although considerable research has focused on flower color in species such as *H. macrophylla*, yet the distinctive patterns of pigment dispersion and scent emission that are still to be investigated. In addition, limited attention has been given to potential links between pigment and volatile biosynthesis pathways in this species; consequently, the mechanisms underlying concurrent petal pigment accumulation and enhanced scent production in ‘PA’ remain elusive.

To elucidate the basis for floral color and fragrance formation in the pink-flowered mutant ‘PA’, we performed corresponding multi-omics analyses—including headspace solid-phase microextraction coupled with gas chromatography–mass spectrometry (HS-SPME-GC-MS) for floral volatiles, ultra-performance liquid chromatography–tandem mass spectrometry (UPLC-MS/MS) for non-volatile metabolome, and RNA-seq—based on data previously obtained for the white-flowered ‘An’ [18]. We systematically compared metabolite profiles and transcript expression levels between the two cultivars, thereby establishing a foundation for deciphering the molecular mechanisms underlying floral scent and color divergence in hydrangea. This research employs a multi-omics approach to offer a comprehensive understanding of the putative associated regulatory mechanism facilitating the molecular-assisted breeding of novel cultivars with improved hue and aroma.

## 2. Materials and Methods

### 2.1. Plant Materials

*H. arborescens* ‘Annabelle’ (An) and *H. arborescens* ‘Pink Annabelle’ (PA) were obtained from the nursery of Kunming Yang Chinese Rose Gardening Co., Ltd. (Kunming, China) for this study. The cultivar ‘An’ and ‘PA’ exhibit inflorescences primarily consisting of sterile flowers, commonly referred to as decorative flowers. The ‘An’ corymb had the pristine white hue of the sterile flower sepals in full bloom, whilst the ‘PA’ displayed a pink coloration of the corymb. Following the collection of inflorescences at full bloom, fresh samples of flowers (comprising sepals and petals) were subjected to headspace solid-phase microextraction coupled with gas chromatography–mass spectrometry (HS-SPME-GC-MS) analysis. A separate portion was immediately excised and flash-frozen in liquid nitrogen prior to storage at −80 °C, and subsequently used for ultra-performance liquid chromatography–tandem mass spectrometry (UPLC-MS/MS) and RNA-seq analyses. All samples were partitioned into three replicas.

### 2.2. RNA Extraction and RNA-Sequencing

Total RNA isolation and RNA sequencing were carried out as described previously [18]. Briefly, total RNA was retrieved employing an RNAprep Pure Plant kit (DP441, Tiangen) followed by Illumina RNA-Seq from Sichuan Panomic Biotechnology Co., Ltd. (Chengdu, China) Magnetic beads with oligo (dT) were utilized to concentrate poly(A) mRNA, which was further fragmented randomly into 200–300 bp. Sequencing adapters were linked to double-stranded cDNAs, and subsequent amplification and purification resulted in the generation of cDNA libraries, which were sequenced employing the Illumina HiSeqTM 2000 system. Library sequencing was carried out from the PANOMIX Biomedical Tech Co., Ltd. (Suzhou, China). The sequencing adapters were trimmed to acquire data of the highest quality, and reads with poor quality containing ≥55 ambiguous bases were discarded using fastp. The GC content of the clean readings was determined, and the Q_20_ and Q_30_ metrics were produced by FastQC to evaluate base quality, subsequently employing Trinity 2.5.1 to compile the high-quality reads. Later on, assembled unigenes were functionally annotated via BLASTX from the GO [40], Pfam data [41], KEGG [42], NCBI [43], and Swiss-Prot [44]. Unigenes’ expression level was quantified employing the FPKM score, whereas DEGs between samples were identified using the R program DESeq version 1.32.0.

### 2.3. UPLC-MS/MS Metabolite Analysis

Metabolite analysis was performed employing UPLC-MS/MS as described earlier [19]. In short, samples were pulverized into a fine powder and obtained using methanol with 4 ppm of 2-Amino-3-(2-chloro-phenyl)-propionic acid following vortexing for 30 s. The supernatant was filtered via a 0.22 μm membrane, transferred to a detection bottle, and analyzed using a Vanquish UHPLC System (Thermo Fisher Scientific, Waltham, MA, USA). Chromatography was performed using an ACQUITY UPLC^®^ HSS T3 (150 2.1 mm, 1.8 μm; Waters Corporation, Shanghai, China). and the column temperature was kept at 40 °C.

### 2.4. Analysis of Floral Volatiles and Volatile Identification

Floral volatile analysis was performed using Headspace solid-phase microextraction (HS-SPME) (Supelco, Bellefonte, PA, USA) coupled to gas chromatography–mass spectrometry (GC-MS) (Agilent, Santa Clara, CA, USA) as explained earlier [33,45]. The flowers were placed in a 200 mL glass vial containing ethyl caprate as the internal standard (1.728 μg), and the vial was sealed with aluminum foil for 15 min. Subsequently, a polydimethylsiloxane (PDMS) fiber was placed in a glass container to absorb volatiles for 15 min prior to injection into an Agilent GC–MS system (Agilent, Santa Clara, CA, USA). The parameters for HS-SPME-GC-MS and metabolite determination were conducted as previously outlined [18,46,47,48]. Volatile substances were identified by comparing them to mass spectra from the NIST Mass Spectral Library (NIST 08) and existing literature. To identify chemicals, mass spectra were matched to NIST 08 with an accuracy of ≥80%. VOCs were identified through homologous n-alkanes (C7-C40; Sigma, St. Louis, MO, USA) extracted under the same GC-MS analytical protocols as previously indicated [35,48]. The relative content of VOCs was calculated with the areas normalized to the internal standard.

### 2.5. Quantitative Real-Time PCR

The qRT-PCR assay was conducted on chosen DEGs linked with the anthocyanin and volatiles production pathways and key transcription factors. Total RNA extraction and cDNA synthesis were performed as previously described [49]. Primer pairs are listed in the Appendix A. Actin was used as an endogenous control. Gene ex-pression levels were determined using the 2^−ΔΔCT^ method [50]. The experiment was performed in triplicate.

### 2.6. Data Analysis

Hierarchical clustering heatmap evaluation was performed using the R package “ComplexHeatmap” (V2.26.0). Orthogonal Projections to Latent Structures-Discriminant Analysis (OPLS-DA) was conducted utilizing the R package “ropls” (V1.41.0), and variable importance in projection (VIP) scores were computed. Supplementary statistical studies, encompassing Pearson correlation analysis, principal component analysis, and Student’s *t*-test, were conducted utilizing R’s built-in features. All omics analyses were performed using three independent biological replicates per cultivar.

## 3. Results

### 3.1. Volatile Organic Compound (VOC) Profiling Between ‘An’ and ‘PA’ Varieties

The GC–MS chromatograms revealed significant differences in the volatile organic compound (VOC) profiles between the white-flowered variety (An) and the pink-flowered variety (PA) of *H. arborescens* (Figure 1A). A total of 25 VOCs were identified in cultivar ‘An’, while 21 VOCs were found in ‘PA’. This indicates that the white variety has slightly greater chemical diversity, but ‘PA’ exhibits a more vigorous emission intensity. The chromatographic peaks for ‘PA’ showed higher relative abundance, suggesting a more active release of volatiles compared to ‘An’.

The identified volatiles primarily included benzenoid/phenylpropanoid derivatives, monoterpenoids, sesquiterpenoids, fatty acid derivatives, and minor other compounds (Figure 1B; Appendix A). Monoterpenoids and benzenoids dominated both varieties, with ‘An’ exhibiting a higher quantity of these compounds compared to ‘PA’. Notably, sesquiterpenoids were absent in ‘PA’. Cultivar ‘An’ displayed elevated levels of aromatic volatiles, such as benzaldehyde, methyl benzoate, and phenylethyl alcohol, which are known contributors to the quality of floral scent.

The Venn diagram (Figure 1C) illustrates the similarities and unique characteristics of the two varieties. Among the detected VOCs, 18 compounds were shared between ‘An’ and ‘PA’, while 5 VOCs (methyl salicylate, isophorone, (+)-4-carene, caryophyllene, and α-farnesene) were unique to ‘An’ and 1 VOC (decanal) was unique to ‘PA’. This overlap indicates that both varieties share a common metabolic foundation for volatile synthesis but differ in their relative contributions to specific compound classes.

Quantitative analysis confirmed that the total relative emission of VOCs was significantly higher in ‘PA’ than in ‘An’ (*p* < 0.01) (Figure 1D). Among the various compound categories, emissions of benzenoid/phenylpropanoids were particularly enriched in ‘PA’ (*p* < 0.01), whereas fatty acid derivatives were found to be more abundant in ‘An’ (*p* < 0.01) (Figure 1E). These findings suggest that although ‘An’ possesses a broader range of VOC species, the ‘PA’ variety produces a higher overall quantity of scent-active aromatic volatiles.

### 3.2. Multivariate and Differential Metabolite Analysis

To investigate the compositional variation in floral volatiles, we performed multivariate statistical analyses employing hierarchical clustering and orthogonal partial least squares discriminant analysis (OPLS-DA) (Figure 2). The heatmap (Figure 2A; Appendix A) clearly delineated the ‘An’ and ‘PA’ samples into two clusters, exhibiting uniform resemblance within groups and significant divergence between them. A considerable quantity of benzenoid and monoterpenoid volatiles, including benzaldehyde, benzyl alcohol, phenylethyl alcohol, (+)-4-carene, β-ocimene, α-pinene, and D-limonene, was detected in greater abundance in ‘PA’ than in ‘An’. In contrast, certain volatiles produced from fatty acids (including isophorone and 2-heptanone) and sesquiterpenes (such as α-farnesene and caryophyllene) were more abundant in ‘An’. Further Student’s *t*-test analysis identified significantly differential compounds. In the more fragrant ‘PA’, seven VOCs showed markedly higher release levels than in ‘An’: five benzenoids/phenylpropanoids (benzaldehyde, benzyl alcohol, acetophenone, α-phenylethyl alcohol, and phenylethyl alcohol), one monoterpenoid (γ-terpinene), and one fatty-acid-derived volatile (decanal). Among these, benzaldehyde, benzyl alcohol, acetophenone, α-phenylethyl alcohol, and phenylethyl alcohol accumulated to 9.58-, 7.95-, 3.74-, 2.50-, and 2.11-fold the levels observed in ‘An’, respectively (Appendix A). These results underscore that the elevated emission of benzenoids/phenylpropanoids (principal floral volatile compounds) likely underlies the stronger fragrance of ‘PA’ compared with ‘An’.

The OPLS-DA score plot (Figure 2B) unequivocally demonstrated a distinct and statistically significant distinction between ‘An’ and ‘PA’, with the first two components representing 52.5% (Component 1) and 30.1% (Component 2) of the total variation, respectively. The close association of biological replicas exhibited robust reproducibility and confirmed the validity of the metabolic data. The VIP analysis (Figure 2C) identified compounds that exert the most substantial influence on varietal discrimination. Significantly, (+)-4-carene, benzyl alcohol, benzaldehyde, α-phenylethyl alcohol, and acetophenone exhibited the highest VIP scores (exceeding 1.0), highlighting their vital role in discriminating the two floral scent profiles (Appendix A). These compounds are predominantly derived from the phenylpropanoid and monoterpenoid biosynthesis pathways, both of which exhibited higher activity in ‘PA’.

In particular, PA accumulates more phenylethyl and benzyl alcohol and expresses more phenylalanine-derived enzymes. Parallel abundance of (+)-4-carene and β-ocimene indicates increased monoterpene synthase activity. These findings suggest that PA’s enhanced fragrance is triggered by coordinated activation of numerous volatile biosynthetic pathways (Appendix A), supporting the metabolic link between floral coloring and aroma intensity.

### 3.3. Comparative Metabolomic Analysis of ‘An’ and ‘PA’ Flowers

A thorough comparative metabolomic analysis of hydrangea flowers was performed using UPLC–MS/MS to clarify the biochemical distinctions between the hydrangea white-flowered variety (An) and its pink-flowered (PA) (Figure 3; Appendix A).

The hierarchical clustering heat map (Figure 3A) distinctly demonstrated a separation between the two varieties, signifying a continuous metabolic divergence. Importantly, the majority of metabolites linked to phenylpropanoid, flavonoid, and terpenoid biosynthesis were observed at increased relative abundances in ‘PA’, while ‘An’ had heightened concentrations of lipid-derived and organic acid-related molecules. This clustering pattern indicates substantial reconfiguration of secondary metabolism associated with the floral color mutation.

The OPLS-DA score plot (Figure 3B) confirmed the distinct metabolic profiles of ‘An’ and ‘PA’, accounting for 68.3% of the total variance in the first two components. The close clustering of biological replicates illustrates the model’s stability and reproducibility, hence enhancing the reliability of metabolite quantification.

A variety of differentially accumulated metabolites (DAMs) between ‘An’ and ‘PA’ were identified, as depicted in the volcano map (Figure 3C; Appendix A). A total of 81 metabolites were significantly elevated and 70 downregulated in ‘PA’ (VIP > 1, *p* < 0.05). The KEGG enrichment analysis (Figure 3D; Appendix A) revealed that the DAMs were predominantly enriched in pathways associated with biosynthesis of phenylpropanoids, phenylpropanoid biosynthesis, linoleic acid metabolism, and Biosynthesis of plant secondary metabolites (Appendix A). The Biosynthesis of phenylpropanoids and Phenylpropanoid biosynthesis exhibited the most substantial enrichment, underscoring their crucial roles in modulating floral color and the synthesis of volatile chemicals. Notably, among the upregulated DAMs in ‘PA’ were metabolites closely linked to floral scent biosynthesis—including benzaldehyde, L-serine, cinnamaldehyde, 2,6-dimethoxyphenol, D-beta-phenylalanine, coumaryl acetate, N-acetyl-D-phenylalanine, and geranyl diphosphate—as well as metabolites associated with anthocyanin or carotenoid biosynthesis, such as cyanidin 3-glucoside, beta-carotene, phenylpyruvic acid, naringenin, eriodictyol, and aesculin (Appendix A). These findings emphasize the positive contribution of these upregulated DAMs to the distinctive floral fragrance and coloration of ‘PA’.

### 3.4. Transcriptome Analysis of ‘An’ and ‘PA’ Flowers

We performed transcriptome sequencing to examine the molecular mechanisms underlying the metabolic divergence, revealing notable discrepancies in gene expression levels (Figure 4). For cultivar ‘An’, the RNA-seq data yielded 94.26 GB of clean reads, with Q_20_ and Q_30_ values of 98.02% and 94.10%, respectively; for cultivar ‘PA’, 94.27 GB of clean reads were generated, with Q_20_ and Q_30_ values of 98.17% and 94.47%, respectively (Appendix A). The PCA score plot (Figure 4A) exhibited a distinct separation between the ‘An’ and ‘PA’ samples along the first main component, which represented a significant fraction of the overall variance. The sample correlation-based cluster heatmap showed that the three biological replicates of each cultivar grouped together, while the two cultivars were distinctly separated (Appendix A). This unique grouping signifies a robust genotype-dependent transcriptional differentiation and good repeatability of the results across biological replicates.

Figure 4B illustrates the overall distribution of differentially expressed genes (DEGs) between the two kinds in the volcano plot. A total of 11,653 DEGs were identified (Appendix A), comprising 7633 upregulated genes and 4020 downregulated genes in ‘PA’ compared to ‘An’ (|log_2_FC| ≥ 1, *p* < 0.05). The elevated genes in ‘PA’ might be primarily linked to the synthesis of secondary metabolites, pigment synthesis, and aroma-related metabolic pathways. Gene Ontology (GO) enrichment analysis (Figure 4C) indicated that the DEGs were significantly enriched in biological processes, including oxidation-reduction process, secondary metabolite biosynthetic process, phenylpropanoid biosynthetic process, etc. These enlarged categories highlight augmented secondary metabolic and regulatory functions in ‘PA’, corresponding with its heightened pigmentation and scent characteristics. Similarly, KEGG pathway enrichment analysis (Figure 4D) identified the significantly enriched pathways (*p* < 0.05), which include phenylpropanoid biosynthesis, flavonoid biosynthesis, and starch and sucrose metabolism. Phenylpropanoid and flavonoid production were prominent routes, directly correlating alterations in gene expression with the observed accumulation of benzyl alcohol, benzaldehyde, and anthocyanin-related metabolites in ‘PA’.

### 3.5. Identification of Structural Biosynthesis DEGs Associated with Secondary Metabolic Pathways

We evaluated the DEGs associated with the synthesis of phenylpropanoids, terpenoids, flavonoids, and anthocyanins to elucidate the molecular basis of the metabolic changes observed in pink-flowered (PA) plants (Figure 5).

In the phenylpropanoid biosynthetic pathway, numerous upstream structural genes, including *PAL* and *4CL*, exhibited considerable upregulation in the ‘PA’ variety compared to the ‘An’ variety (Appendix A). These enzymes facilitate the first reactions that convert phenylalanine into p-coumaroyl-CoA, an essential precursor for the synthesis of both lignin and benzenoids. In both cultivars, *PAL* (TRINITY_DN1793_c4_g1 and TRINITY_DN56496_c0_g1), *4CL* (TRINITY_DN4000_c0_g1), phenylpyruvate decarboxylase (*PPDC*) (TRINITY_DN7239_c0_g1), *DFR* (TRINITY_DN4175_c0_g1), and two benzoic acid/salicylic acid carboxyl methyltransferase (*BSMT*) genes (TRINITY_DN898_c0_g1 and TRINITY_DN898_c1_g2) were found, exhibiting distinct expression patterns. The *PAL*, *PPDC* and *BSMT* genes exhibited increased expression in cultivar ‘PA’, whereas *4CL* demonstrated a higher level of expression in cultivar ‘AN’ (Appendix A).

In the terpenoid biosynthesis pathway, the *GGPS* (geranylgeranyl diphosphate synthase) and *TPS* (terpene synthase) genes are essential for terpenoid formation. *TPS* genes transform GPP and FPP into monoterpenoids and sesquiterpenoids, respectively. We identify three *TPS* (TRINITY_DN16935_c1_g1, TRINITY_DN8448_c0_g1, and TRINITY_DN18724_c0_g1) and two *GGPS* (TRINITY_DN3548_c0_g1 and TRINITY_DN556_c7_g1) from the dataset. The two *GGPS* and two *TPS* genes (excluding TRINITY_DN16935_c1_g1) exhibited the higher expression in the ‘PA’ cultivar, which demonstrates an elevated level of VOC, signifying their essential function in VOC synthesis.

Cultivar ‘PA’ showed high transcriptional activity throughout the flavonoid and anthocyanin production pathway. Key enzymes such as *C4H*, *CHS*, *ANS*, *F3′H*, *ANR*, *DFR*, *BZ1*, *F3′5′H*, *UGT75C1* (UDP-glycosyltransferases), and *FLS* (flavonol synthase) were significantly overexpressed in ‘PA’ compared to ‘An’. Among them, *C4H* (TRINITY_DN28015_c0_g1), *CHS* (except TRINITY_DN35870_c0_g2), *ANS* (TRINITY_DN18999_c0_g1), *FLS* (TRINITY_DN10659_c0_g1), *ANR* (TRINITY_DN3485_c0_g1), *DFR* (TRINITY_DN6477_c0_g1), *F3′5′H* (TRINITY_DN23393_c0_g1), *UGT75C1* (TRINITY_DN2020_c1_g1) and *BZ1* (TRINITY_DN9567_c0_g1) had highest expression level in cultivar ‘PA’, highlighting their putative role in floral coloration. The elevated expression level of these genes might be directly involved in the synthesis and glycosylation of anthocyanins, resulting in the distinctive pink coloration of ‘PA’ petals.

### 3.6. Transcription Factor Associated with Secondary Metabolism

To further comprehend the regulatory mechanisms underlying transcriptional modification in PA flowers, we analyzed transcription factor [51] families among DEGs (Figure 6; Appendix A). Our data demonstrated that 36 TF families showed significant expression changes between ‘An’ and ‘PA’, indicating comprehensive transcriptional regulation underlying secondary metabolism.

Significantly, transcription factor families including *ERF* (12 upregulated and 1 downregulated), *MYB* (11 upregulated and 2 downregulated), *bZIP* (8 upregulated and 2 downregulated), *WRKY* (6 upregulated), *NAC* (9 upregulated and 1 downregulated), and *C2H2* (10 upregulated and 6 downregulated) were primarily elevated in ‘PA’, suggesting their probable involvement in the activation of genes related to phenylpropanoid, flavonoid, and terpenoid biosynthesis pathways. *MYB* and *bHLH* factors are determined mediators of anthocyanin biosynthesis, collaborating to enhance pigment accumulation by triggering structural genes like *CHS*, *DFR*, and *ANS*. Overall, most of the transcription factors (100) were upregulated, while 35 were downregulated in the pink flowering cultivar compared to ‘An’ (Appendix A).

### 3.7. Verification of DEGs Profiling by qRT-PCR

To validate the RNA-seq results, ten differentially expressed genes associated with floral color and aroma production were randomly selected for qRT-PCR investigation (Appendix A). The randomly selected genes, and the transcription factors for validation were (TRINITY_DN18999_c0_g1 (*ANS*), TRINITY_DN3485_c0_g1 (*ANR*), TRINITY_DN2020_c1_g1 (*UGT75C1*), TRINITY_DN9567_c0_g1 (*BZ1*), TRINITY_DN8448_c0_g1 (*TPS*), TRINITY_DN18724_c0_g1 (*TPS*), TRINITY_DN3715_c0_g1 (*ERF*), TRINITY_DN15566_c0_g1 (*MYB*), TRINITY_DN739_c0_g3 (*NAC*) (Figure 7). The results demonstrated that the expression profiles of these genes, assessed via qRT-PCR, closely aligned with the RNA-seq data values. The expression levels of *ANS*, *UGT75C1*, *BZ1*, and *ANR* were highest in the cultivar ‘PA’ compared to ‘An’. The expression levels of *ERF* were highest in ‘An’, while *MYB* and *NAC* exhibited maximal mRNA transcript levels in ‘PA’. The qRT-PCR experiments confirmed the reliability of the RNA-seq-derived gene expression profiles.

## 4. Discussion

The current study offers a comprehensive molecular understanding of how floral color and fragrance are co-regulated in *H. arborescens* ‘Annabelle’. By integrating volatile profiling, metabolomics, and transcriptomics, we demonstrate that the pink-flowered (PA) achieves heightened pigmentation and aroma through the coordinated activation of the phenylpropanoid, flavonoid, and terpenoid pathways. These pathways work together to synthesize anthocyanins and volatile aromatic compounds, highlighting a common biochemical and regulatory basis for the diversity of floral color and scent.

### 4.1. Versatile Pathways of Floral Volatiles Biosynthesis in ‘An’ and ‘PA’

The gas chromatography-mass spectrometry (GC–MS) data analysis revealed significant quantitative and qualitative differences in the VOC profiles of both cultivars ‘An’ and ‘PA’. We found that ‘An’ produced a somewhat greater diversity of VOC species (25 vs. 21); nevertheless, ‘PA’ emitted substantially more total volatiles. This increased emission was mainly attributed to the prominent high level of benzenoid/phenylpropanoid and monoterpenoid compounds, indicating an elevated metabolic flow across both the phenylalanine-derived and MEP pathways. The main volatile compounds identified in ‘PA’ include benzaldehyde, benzyl alcohol, phenylethyl alcohol (2-PE), and methyl benzoate. These compounds are characteristic benzenoid aromatics that originate from L-phenylalanine through the phenylpropanoid metabolic pathway mediated by phenylalanine ammonia-lyase (PAL) [31,52].

These results align with earlier studies on *Petunia hybrida* and *Rosa rugosa*, where elevated levels of *PAL* and *BSMT* promoted the formation of benzenoid volatiles that markedly improve floral scent [29,53]. The higher amounts of benzyl alcohol and phenylethyl alcohol in hydrangea ‘PA’ indicate an enhanced activity of phenylacetaldehyde synthase (PAAS) and phenylacetaldehyde reductase (PAR), enzymes that transform phenylalanine into volatile alcohols that contribute to the distinctive sweet floral fragrance [54]. The abundance of methyl benzoate and methyl salicylate in ‘PA’ signifies higher expression of *BSMT*, hence underlining its crucial role in the biosynthesis of floral aroma. Likewise, higher emissions of monoterpenes, such as β-ocimene, (+)-4-carene, and α-pinene, indicate an upregulation of terpenoid biosynthesis, possibly facilitated by the functions of geranyl diphosphate synthase (GPPS) and terpene synthase [55]. The simultaneous accumulation of benzenoids and monoterpenes indicates a cohesive regulatory network regulating both the shikimate and isoprenoid pathways, similar to those observed in *Clarkia breweri* and *Nicotiana tabacum*, where the transcriptional activation of *TPS* genes is associated with stronger floral scent emission [22,23,56,57].

The cultivar ‘An’ exhibited elevated levels of fatty acid-derived volatiles, including isophorone and 2-heptanone, indicating a distinct allocation of carbon precursors between the two cultivars. The metabolic shift from lipid-derived volatiles in ‘An’ to aromatic and terpenoid volatiles in ‘PA’ signifies a purposeful reallocation of resources that enhances the synthesis of powerful floral fragrance compounds.

### 4.2. Metabolomic Reconfiguration and Secondary Metabolite Enrichment

Combined UPLC–MS/MS metabolomic analysis validated the significant metabolic divergence between the two varieties. ‘PA’ exhibited significant enrichment of metabolites connected with phenylpropanoid, flavonoid, and monoterpenoid pathways, whereas ‘An’ accumulated elevated quantities of lipid-derived and organic acid-related molecules. Significantly, 81 metabolites were upregulated and 70 were downregulated in ‘PA’, indicating a substantial reallocation of carbon flow towards pigment- and fragrance-related biosynthesis.

Benzyl alcohol, acetophenone, quercetin, and naringenin emerged as prominent intermediates in the metabolism of phenylalanine and tyrosine, serving as precursors for anthocyanin and benzenoid production. KEGG enrichment analysis verified that biosynthesis of phenylpropanoids and phenylpropanoid biosynthesis were among the most significantly regulated pathways (*p* < 0.01), corresponding with the elevated levels of color and scent metabolites. The results align with observations in *Petunia*, *Camellia*, and *Rhododendron*, wherein the activation of these pathways simultaneously enhances floral pigmentation and fragrance strength [31,58,59,60]. Consequently, the metabolic evidence succinctly suggests that the floral color modification in ‘PA’ might have triggered a coordinated metabolic reprogramming, reallocating resources toward secondary metabolism, which simultaneously amplifies pigmentation and volatile production.

### 4.3. Transcriptomic Regulation of Pigment and Volatile Organic Compound Biosynthesis

RNA-Seq dataset revealed 11,653 DEGs between both cultivars, with 7633 upregulated in ‘PA’, many of which may be associated with phenylpropanoid and flavonoid production. PAL, C4H, and 4CL are crucial enzymes that catalyze the transformation of L-phenylalanine into p-coumaroyl-CoA, the principal precursor for flavonoid and benzenoid pathways, and demonstrated substantial overexpression in ‘PA’. Likewise, elevated expression of *CHS*, *CHI*, *F3′H*, *DFR*, *ANS*, and *UFGT*/*BZ1* aligns with higher anthocyanin biosynthesis, correlating with the amplified pink coloring noted in ‘PA’ sepals.

The terpenoid pathway exhibited increased expression of *GPPS* and *TPS*, which correlated with heightened emission of monoterpenes, including β-ocimene and α-pinene. Comparable transcript–metabolite coupling between *GPPS*/*TPS* expression and VOC production has been established in *Clarkia breweri* and *Nicotiana* species [23,57,61,62]. The simultaneous elevation of PAL–C4H–4CL, CHS–DFR–ANS, PAAS/PAR–BSMT, and GPPS/TPS delineates a molecular pathway that directs metabolic flux from primary precursors into color and aroma pathways, elucidating the coordinated improvement of visual and olfactory features in the ‘PA’ flowers. Although, the data offer novel insights and a foundation for future research; yet, validation through molecular assays is required.

Thirty-six transcription factor families demonstrated differential expression, including MYB, bHLH, ERF, and WRKY. The MYB–bHLH–WD40 (MBW) complex is pivotal in regulating anthocyanin biosynthesis by directly activating *CHS*, *DFR*, and *ANS* [63,64,65,66]. Simultaneously, the ERF and WRKY families enhance benzenoid and terpenoid biosynthesis through the upregulation of *PAL*, *4CL*, *BSMT*, and *TPS* [49,50,65,67,68].

### 4.4. Coordinated Color–Scent Metabolic Interaction

Coordinated variation in floral scent and color represents an important adaptive strategy in plant ecological interactions. The coordinated regulation of floral scent and floral color relies not only on shared metabolic pathways but also on distinct gene regulatory [6,69]. Previous studies have revealed two major biosynthetic connections between scent and pigmentation. First, anthocyanin pigments (responsible for blue, purple, and red coloration) and volatile phenylpropanoid compounds both derive from the phenylpropanoid pathway [70]. Secondly, carotenoid pigments (yellow, orange, and red) and volatile isoprenoids/carotenoid-derived volatiles are synthesized through the plastidial 2-C-methyl-D-erythritol-4-phosphate (MEP) pathway [71,72,73].

The substantial evidence from the VOC, metabolomic, and transcriptome analyses reveals that the superior pigmentation and scent of ‘PA’ might originate from a system-wide reprogramming of phenylalanine- and isoprenoid-derived secondary metabolism. At the metabolite level, hydrangea ‘PA’ shows marked accumulation of benzenoids/phenylpropanoids (e.g., benzaldehyde, benzyl alcohol, α-phenylethyl alcohol, phenylethyl alcohol) and anthocyanin-related compounds (e.g., cyanidin 3-glucoside, naringenin), reflecting the transcriptional activation of their biosynthetic enzymes. Simultaneously, elevated levels of anthocyanin-related metabolites (e.g., naringenin, quercetin) correspond with the activation of flavonoid structure genes, affirming a close coordination between metabolic and transcriptional mechanisms. We propose that the phenylpropanoid pathway serves as the metabolic nexus integrating pigment and aroma biosynthesis via different interrelated axes in pink-flowered hydrangea: precursor amplification, where the overexpression of PAL–C4H–4CL enhances the flux from phenylalanine to p-coumaroyl-CoA, hence driving the formation of both anthocyanins and benzenoids. Secondly, branch-specific activation of CHS–DFR–ANS–UFGT/BZ1 enhances anthocyanin accumulation (color), whereas PAAS–PAR–BSMT stimulate the formation of benzenoids/phenylpropanoids VOCs. Thirdly, the regulatory enhancement of transcription factors (MYB, bHLH, ERF, and WRKY) families guarantees coordinated transcriptional regulation of pigment and volatile biosynthesis. Similarly, in *Rosa damascena*, during three stages of floral development, the highly expressed *RhPAR* gene induces the upregulation of *RhMYB1* and *RhANS*, thereby enhancing the production of the phenylpropanoid volatile 2-phenylethanol as well as anthocyanin accumulation (purple-red pigmentation) [58,74,75]. Floral scent and color are also correlated in *Papaver nudicaule*: in yellow-flowered cultivars, emission of the volatile indole is significantly elevated, whereas this pattern is absent in orange-flowered individuals [76].

However, floral fragrance and floral coloration do not always exhibit synergistic accumulation. Studies in *Dianthus caryophyllus* have shown that suppression of structural genes in the anthocyanin biosynthetic pathway reduces floral pigment accumulation, accompanied by increased emission of phenylpropanoid volatiles such as methyl benzoate [77]. In *Brassica napus*, functional mutations in carotenoid cleavage dioxygenase (CCD) convert fragrant white flowers into non-fragrant yellow flowers [78]. In short, ‘PA’ defines a model of metabolic and transcriptional synergy, wherein the phenylpropanoid pathway might serve as a central hub orchestrating precursor flux, branch activation, and transcription factor-mediated regulation to yield an interconnected floral characteristic differentiated by enhanced color and fragrance.

## 5. Conclusions

An integrative investigation of volatile compounds, metabolite profiles, and transcriptome expression elucidated a coordinated molecular mechanism responsible for the enhanced floral aroma and pigmentation in the pink-flowered *Hydrangea arborescens* ‘Pink Annabelle’ (PA) relative to the white-flowered *H. arborescens* ‘Annabelle’ (An). ‘PA’ exhibited increased emission of benzenoid and phenylpropanoid volatiles, elevated accumulation of flavonoids and anthocyanins, and notable activation of structural genes in the phenylpropanoid and flavonoid biosynthesis pathways. Transcriptomic analysis revealed considerable expression of *PAL*, *4CL*, *CHS*, *DFR*, and *ANS*, triggered by transcription factors including *MYB*, *bHLH*, *ERF*, and *WRKY*. The molecular and metabolic modifications elucidate the enhanced floral color and aroma of the ‘PA’ variety, underscoring the synchronized regulation of scent-and pigment-associated secondary metabolism in hydrangea flowers.

## Figures and Tables

**Figure 1 plants-15-00155-f001:**
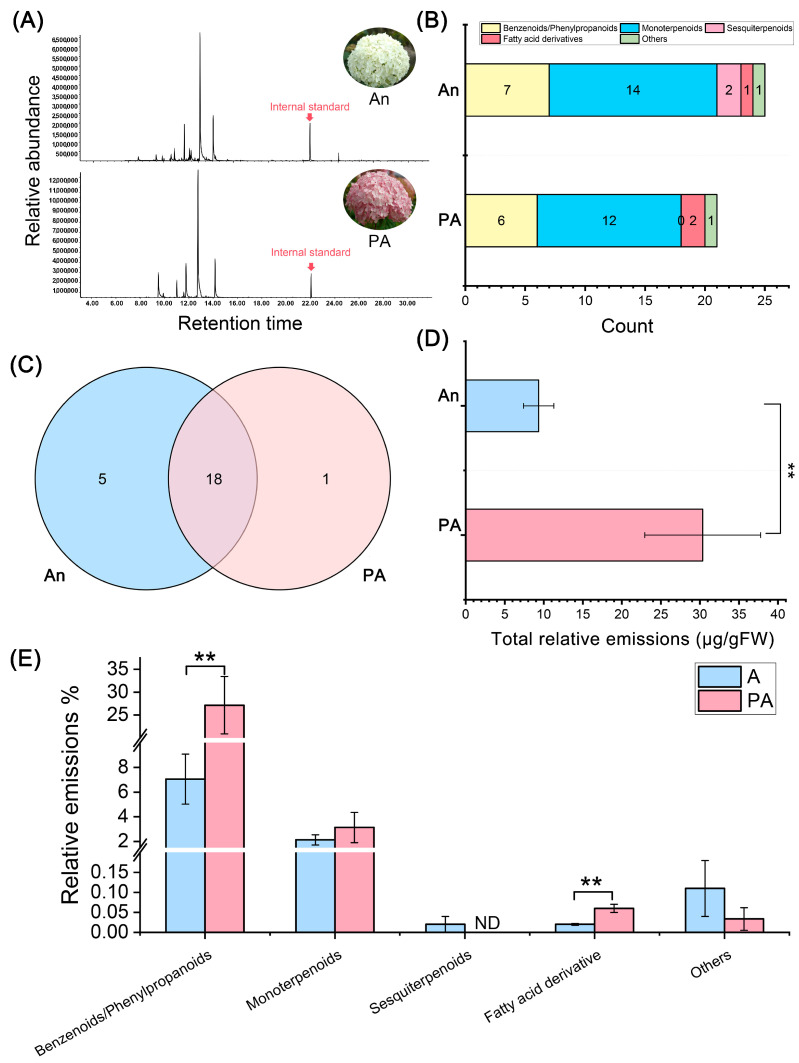
Detection and Analysis of volatile organic compounds (VOCs) in ‘An’ and its ‘PA’ floral scent using HS-SPME-GC-MS. (**A**) A representative total ion chromatogram (TIC). (**B**) Comparative analysis of the number of VOCs categorized across different groups. (**C**) Venn diagram analysis, (**D**) assessment of the total VOCs released, and (**E**) comparative analysis of VOC emissions in two distinct hydrangea cultivars. ** Statistics indicate significance at *p* < 0.01.

**Figure 2 plants-15-00155-f002:**
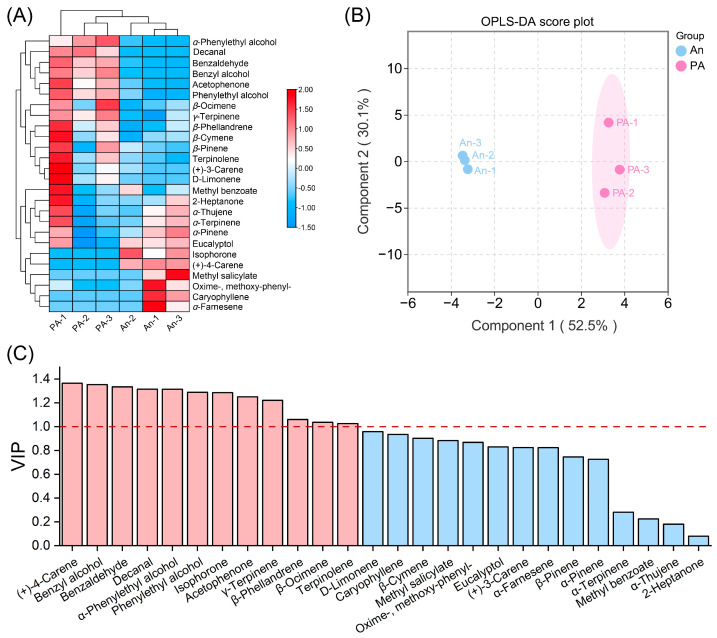
Multivariate statistical investigation of floral volatile organic compound emissions from ‘An’ and ‘PA’. (**A**) Heatmap based on hierarchical clustering of normalized VOC emission values. (**B**) OPLS-DA score plot, and (**C**) VIP value comparison for 26 VOCs. In the VIP analysis, ‘AN’ were compared to the ‘PA’ variants, with values exceeding 1 indicating the primary signature compounds.

**Figure 3 plants-15-00155-f003:**
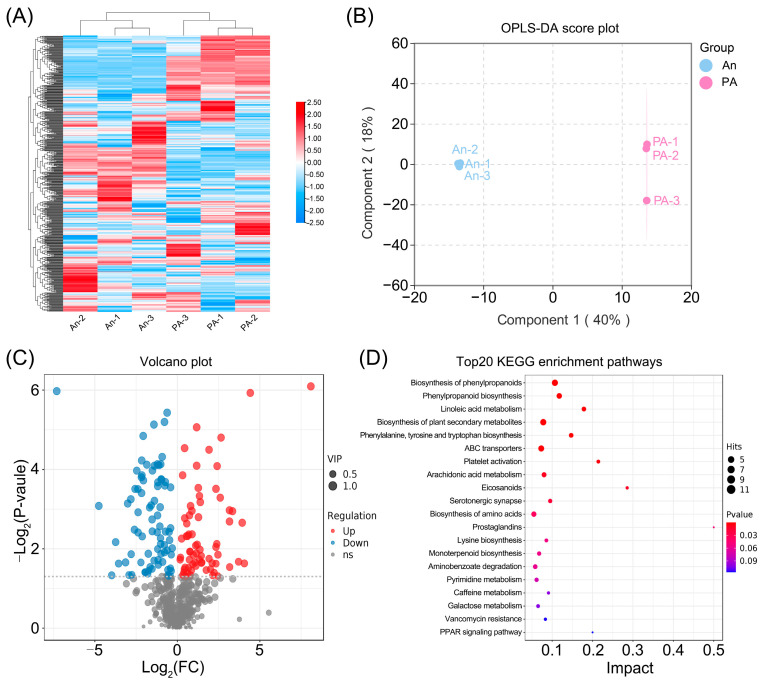
Comparative study of metabolites in An and PA flowers employing UPLC-MS/MS detection. (**A**) Heatmap of hierarchical clustering based on normalized gene FPKM values. (**B**) OPLS-DA score plot, (**C**) Volcano plot assessment, and (**D**) KEGG enrichment analysis of differential metabolites.

**Figure 4 plants-15-00155-f004:**
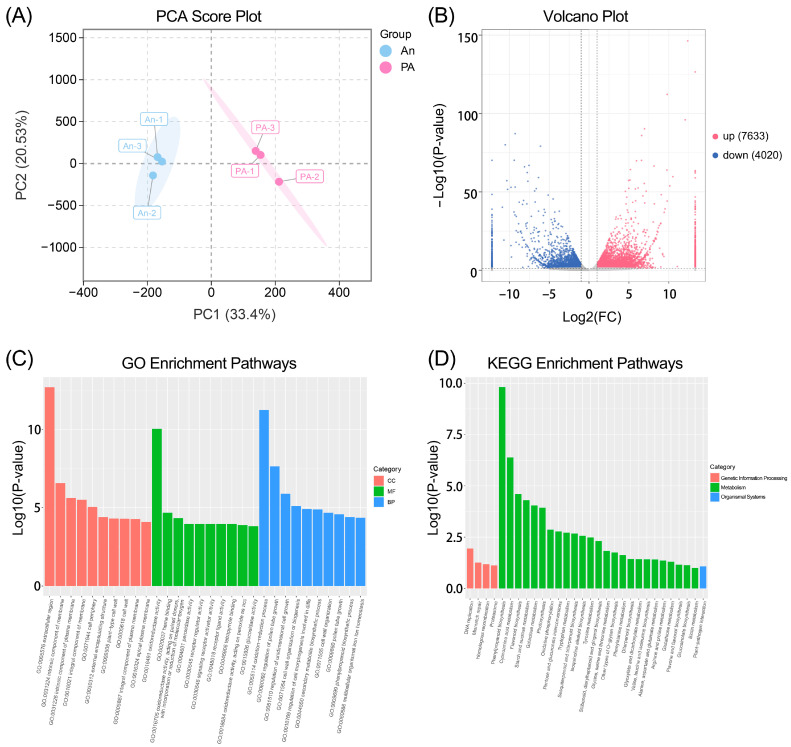
Transcriptomic analysis of An and PA flowers. (**A**) PCA score plot, (**B**) volcano plot, (**C**) gene ontology enrichment, and (**D**) Kyoto Encyclopedia of Genes and Genomes enrichment analysis of DEGs.

**Figure 5 plants-15-00155-f005:**
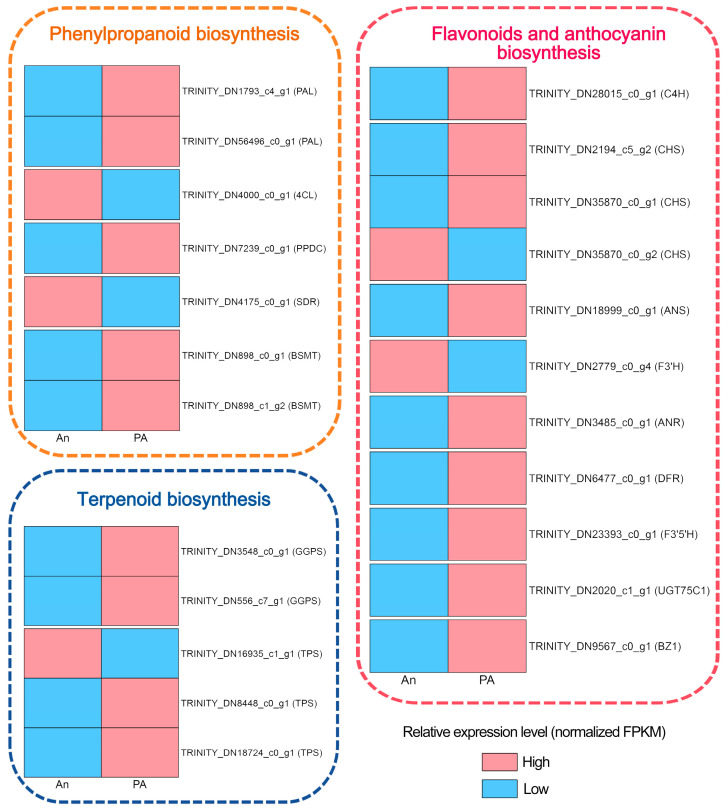
Differentially expressed genes in phenylpropanoid biosynthesis, terpenoid biosynthesis, flavonoid biosynthesis, and anthocyanin biosynthesis.

**Figure 6 plants-15-00155-f006:**
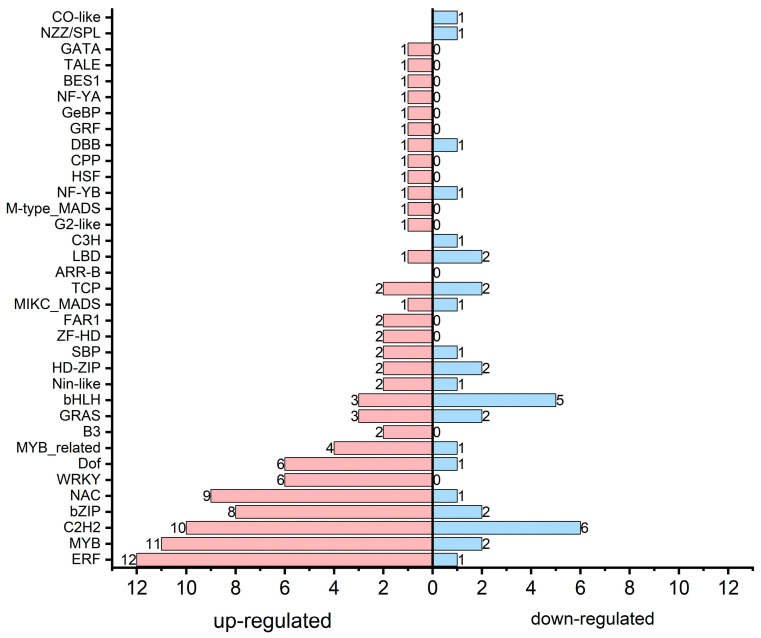
Transcription factors identified among the DEGs.

**Figure 7 plants-15-00155-f007:**
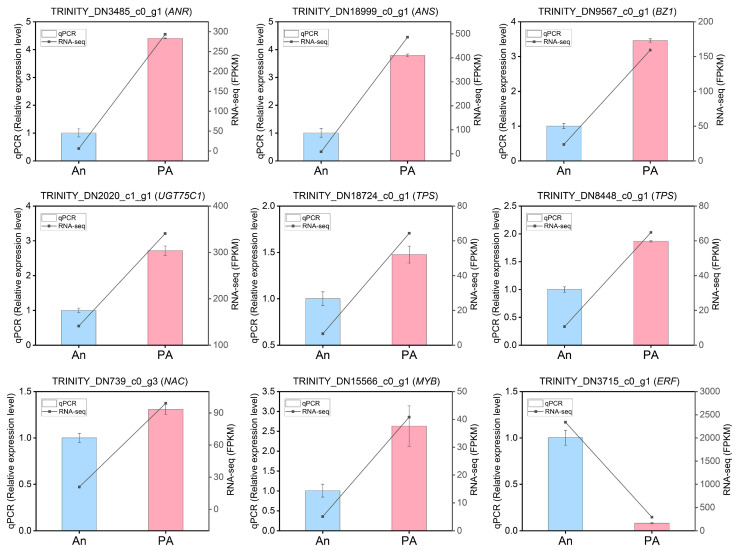
Validation of the expression profiles of ten randomly selected genes using qRT-PCR in together with RNA-seq data. Actin was used as an endogenous control. Data is denoted as the mean ± SEM (n = 3).

## Data Availability

The raw RNA-seq data are available at the NCBI database (PRJNA954056). The data that supports the findings of this study are available in the Supporting Information of this article.

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
