# Peer review of "Elucidating Scent and Color Variation in White and Pink-Flowered Hydrangea arborescens ‘Annabelle’ Through Multi-Omics Profiling"

_plants, 2026, doi:10.3390/plants15010155_

Round 1

Reviewer 1 Report

Comments and Suggestions for Authors

COMMENTS TO THE AUTHORS

GENERAL COMMENT

The manuscript presents a well-executed multi-omics study integrating volatile profiling, metabolomics, and transcriptomics to investigate floral color and scent variation in Hydrangea arborescens. The datasets are robust, internally consistent, and well supported by the Supplementary Materials. The study is relevant to ornamental plant biology and breeding.

Some conclusions, however, would benefit from more cautious wording, particularly where correlative multi-omics evidence is discussed in terms of coordinated or causal regulatory mechanisms.

MAJOR COMMENTS

Interpretation of multi-omics integration
In several sections (Abstract; Results Section 3.2; Discussion Sections 4 and 4.4), the manuscript implies coordinated or synchronized regulatory mechanisms linking floral color and scent. While the data clearly demonstrate strong associations between gene expression and metabolite accumulation, functional validation is not provided.
Suggestion: Please moderate causal language and clearly distinguish correlation from causation.

RNA-seq validation
Key structural genes involved in phenylpropanoid, flavonoid, and terpenoid biosynthesis are extensively discussed (Results Sections 3.4–3.6; Supplementary Tables S8–S10), but no independent validation (e.g., qRT-PCR) is presented.
Suggestion: Please explicitly acknowledge this limitation and clarify the exploratory nature of the transcriptomic analysis.

Biological replicates
Three biological replicates are mentioned, and the Supplementary Materials support good reproducibility.
Suggestion: Please explicitly state that all omics analyses were performed using three independent biological replicates per cultivar.

MINOR COMMENTS

-Please use consistent terminology for the pink-flowered cultivar (either “PA” or “Pink Annabelle”).

-Clarify whether VOC data are expressed as relative abundances or semi-quantitative values (Methods Section 2.4; Results Section 3.1).

-Improve the readability of heatmaps in Figures 2 and 3.

Author Response

Reviewer 1

COMMENTS TO THE AUTHORS

GENERAL COMMENT

The manuscript presents a well-executed multi-omics study integrating volatile profiling, metabolomics, and transcriptomics to investigate floral color and scent variation in Hydrangea arborescens. The datasets are robust, internally consistent, and well supported by the Supplementary Materials. The study is relevant to ornamental plant biology and breeding.

Some conclusions, however, would benefit from more cautious wording, particularly where correlative multi-omics evidence is discussed in terms of coordinated or causal regulatory mechanisms.

MAJOR COMMENTS

Interpretation of multi-omics integration
In several sections (Abstract; Results Section 3.2; Discussion Sections 4 and 4.4), the manuscript implies coordinated or synchronized regulatory mechanisms linking floral color and scent. While the data clearly demonstrate strong associations between gene expression and metabolite accumulation, functional validation is not provided.
Suggestion: Please moderate causal language and clearly distinguish correlation from causation.

Response: Dear reviewer, the amendments have been made as per your insightful suggestion.

RNA-seq validation
Key structural genes involved in phenylpropanoid, flavonoid, and terpenoid biosynthesis are extensively discussed (Results Sections 3.4–3.6; Supplementary Tables S8–S10), but no independent validation (e.g., qRT-PCR) is presented.
Suggestion: Please explicitly acknowledge this limitation and clarify the exploratory nature of the transcriptomic analysis.

Response: Dear reviewer, we performed qRT-PCR analysis to validate the data as instructed, as well as highlighted the imitations of current data.

Biological replicates
Three biological replicates are mentioned, and the Supplementary Materials support good reproducibility.
Suggestion: Please explicitly state that all omics analyses were performed using three independent biological replicates per cultivar.

Response: Dear reviewer, the amendments have been made as instructed.

MINOR COMMENTS

-Please use consistent terminology for the pink-flowered cultivar (either “PA” or “Pink Annabelle”).

Response: The term PA is consistent throughout the text.

-Clarify whether VOC data are expressed as relative abundances or semi-quantitative values (Methods Section 2.4; Results Section 3.1).

Response: the data is expressed as relative abundance and we have added a sentence in the methodology section.

-Improve the readability of heatmaps in Figures 2 and 3.

Response: Dear reviewer, the heatmap figures are provided separately and it’s up to journal standard with resolution more than 300 dpi.

Reviewer 2 Report

Comments and Suggestions for Authors

This study comprehensively investigated the differences in color and fragrance between white and pink Hydrangea flowers using integrated metabolome and transcriptome analyses. While these data are expected to be useful for future Hydrangea breeding, many of the presented data are in the form of heatmaps, making it difficult for readers to grasp the actual accumulation levels of substances and gene expression levels.

For key pigments and volatile components, the discussion should go beyond merely mentioning up-regulation or down-regulation between white and pink flowers based on heatmaps, and should include the actual accumulation/expression levels and their statistical significance. Although the transcriptome data is shown with three replicates for each sample, statistical significance tests are not performed even in the supplementary data, making it unclear whether the differences are genuine. Furthermore, understanding the content may be difficult for readers unfamiliar with the metabolic pathways of volatile compounds and pigments; therefore, I recommend including diagrams of the metabolic pathways in the supplementary data.

The information would be more valuable if the discussion elaborated more specifically on which volatile components are the primary substances in Hydrangea and which enzyme genes are crucial for their biosynthesis. Similarly, the discussion should specify which pigment substances are key and which enzyme genes are important for their synthesis. Currently, the results presented are merely the outcome of the omics analyses, and the biological significance of these findings specifically for Hydrangea is not easily conveyed to the reader.

Other minor points:

  1. 55: The term " phenylalanine ammonia-lyase (PAL)" is incorrect.
  2. 56: Add the abbreviation, "chalcone synthase (CHS)".
  3. 58: The phrase "colorless anthocyanins" is unclear; please clarify what "colorless" intends to mean here.
  4. 62: The "-O-" in "flavonoid 3-O-glucosyltransferase" should be italicized.
  5. 100: The cultivar name "pink annabelle" should be unified to "Pink Annabelle" throughout the manuscript and elsewhere.
  6. 177: It is unclear what value the "variable importance in projection (VIP)" represents and how white flowers were compared to pink flowers. An explanation is necessary for readers to interpret the figure.
  7. 287: Is "94.26 clean read" meant to be "94.26% clean read"? Please specify the unit and check other related entries.
  8. 328-330: The abbreviations such as BSMT and PALPPDC are undefined. The full official name must be provided when the term appears in the text for the first time.

Author Response

Reviewer 2

Comments and Suggestions for Authors

This study comprehensively investigated the differences in color and fragrance between white and pink Hydrangea flowers using integrated metabolome and transcriptome analyses. While these data are expected to be useful for future Hydrangea breeding, many of the presented data are in the form of heatmaps, making it difficult for readers to grasp the actual accumulation levels of substances and gene expression levels.

Response:

Dear reviewer, we appreciate your useful suggestion, yet the data of volatiles and transcriptome is too big to shown in graph, therefore we utilize heatmap to represent for convenience. For future studies, we will keep it in my mind and draw accordingly following your kind suggestion.

For key pigments and volatile components, the discussion should go beyond merely mentioning up-regulation or down-regulation between white and pink flowers based on heatmaps, and should include the actual accumulation/expression levels and their statistical significance. Although the transcriptome data is shown with three replicates for each sample, statistical significance tests are not performed even in the supplementary data, making it unclear whether the differences are genuine. Furthermore, understanding the content may be difficult for readers unfamiliar with the metabolic pathways of volatile compounds and pigments; therefore, I recommend including diagrams of the metabolic pathways in the supplementary data.

Response:

Dear reviewer, we have made modification in the text and figure as per your suggestion. The statistical analysis on volatiles has been performed and are highlighted in Table S1. Furthermore, in the results and discussion section we have performed modification with statistical data analysis. Also, included the metabolic figure in the supplemental data. Moreover, the figures are statistical analyzed and we added the significant and non-significant replicates.

The information would be more valuable if the discussion elaborated more specifically on which volatile components are the primary substances in Hydrangea and which enzyme genes are crucial for their biosynthesis. Similarly, the discussion should specify which pigment substances are key and which enzyme genes are important for their synthesis. Currently, the results presented are merely the outcome of the omics analyses, and the biological significance of these findings specifically for Hydrangea is not easily conveyed to the reader.

 Response:

Dear reviewer, following your valuable suggestion, we have incorporated the discussion highlighting key volatiles and biosynthesis genes associated with metabolic pathway.

Other minor points:

  1. 55: The term " phenylalanine ammonia-lyase (PAL)" is incorrect.

Response: Corrected, thank you for highlighting.

  1. 56: Add the abbreviation, "chalcone synthase (CHS)".

Response: The term is highlighted on page 3-line number 345

  1. 58: The phrase "colorless anthocyanins" is unclear; please clarify what "colorless" intends to mean here.

Response: Dear reviewer we have provided the explanatory sentences about the colorless anthocyanin as per your kind suggestion.

  1. 62: The "-O-" in "flavonoid 3-O-glucosyltransferase" should be italicized.

Response: Modified as per instructed.

  1. 100: The cultivar name "pink annabelle" should be unified to "Pink Annabelle" throughout the manuscript and elsewhere.

Response: The name Pink Annabelle is unified now throughout the text.

  1. 177: It is unclear what value the "variable importance in projection (VIP)" represents and how white flowers were compared to pink flowers. An explanation is necessary for readers to interpret the figure.

Response: Dear reviewer, we appreciate for highlighting key points to improve the manuscript. Following your valuable suggestion we have incorporated the explanatory sentence in the text at corresponding section.

  1. 287: Is "94.26 clean read" meant to be "94.26% clean read"? Please specify the unit and check other related entries.

Response: Dear reviewer, it was 94.26 GB, we have added in the text.

  1. 328-330: The abbreviations such as BSMT and PAL PPDC are undefined. The full official name must be provided when the term appears in the text for the first time.

Response: The missing abbreviations are added in the text.